

# The efficacy of Virkon-S for the control of saprolegniasis in common carp, *Cyprinus carpio* L

Haitham Saeed Rahman[1] and Tae-Jin Choi[2]

[1] KOICA-PKNU International Graduate Program of Fisheries Science, Pukyong National University, Busan, South Korea

[2] Department of Microbiology, Pukyong National University, Busan, South Korea

## ABSTRACT

**Background**. *Saprolegnia parasitica* is a fish pathogen that causes severe economic losses worldwide. Virkon-S is a well-known disinfectant known to exhibit antimicrobial activities against bacteria, viruses, and fungi. In this study, we tested the anti-fungal activity of Virkon-S against *S. parasitica*, the major causal agent of saprolegniasis.

**Methods**. The lowest concentration of Virkon-S that prevented germination or the visible growth of spores and the percent spore germination were determined using potato dextrose agar plates containing different concentrations of Virkon-S. The cytotoxic effect was evaluated using the Ez-Cytox Cell Viability Assay with epithelioma papulosum cyprini (EPC) cells grown in L-15 medium and acute toxicity tests were carried out with cultured fingerlings of common carp for 96 h. Artificial infection with *S. parasitica* was performed by placing the fish in tanks containing zoospores of *S. parasitica* after descaling and wounding at three positions. The diseased fish were kept in tanks containing 2, 4, and 10 ppm of Virkon-S for 10 days to observe the treatment effect.

**Results**. The *in vitro* assay results showed that Virkon-S could inhibit spore germination and the resulting mycelial growth at a concentration as low as 4 ppm. No cytotoxic effect on EPC cells was observed even at a concentration as high as 100 ppm. Additionally, no acute toxicity in the common carp was observed at 10 ppm following 96 h exposure. Ten days of treatment with 4 and 10 ppm Virkon-S resulted in complete reversal of artificially-induced saprolegniasis in the common carp.

**Discussion**. This data indicates that Virkon-S can be used for the control of saprolegniasis without harmful effects in fish. However, further research on the effect in humans and food supplies is necessary.

Corresponding author
Tae-Jin Choi, choitj@pknu.ac.kr

## INTRODUCTION

Fungal infections are one of the main factors of mortality and economic loss among the ornamental and food fish farming industries (*Fregeneda-Grandes, Rodríguez-Cadenas & Aller-Gancedo, 2007*; *Jalilpoor, Shenavar Masouleh & Masoumzadeh, 2006*). The most common and economically important fungal disease of cultured fish is saprolegniasis. Saprolegniasis, which is also known as 'winter fungus', usually occurs between October

and March when the water temperature is below 15 °C, but mortality usually increases as the temperature rises in early spring (*Osman et al., 2008*). *Saprolegnia* infections are visible to the naked eye as white patches on the skin of the infected fish or as 'cotton wool' on fish eggs. From these eggs the fungus can spread to live eggs via positive chemotaxis, meaning some chemical signals from the live eggs cause the fungus to move towards them (*Bruno & Wood, 1999*).

The term 'saprolegniasis' describes infection with fungi, actually a phylogenetic lineage of fungus-like microorganisms of the family *Saprolegniaceae* in the order *Saprolegniales* of class *Oomycota*. Two main genera, *Saprolegnia* and *Achlya* of the family *Saprolegniaceae*, can infect fish or shellfish. Although the pathogenic organisms responsible for saprolegniasis have not yet been identified, three species including *Saprolegnia parasitica* and *S. diclina* which are taxonomically difficult group and referred as the *Saprolegnia* complex (*Willoughby, Pickering & Johnson, 1984*), and *Achlya hoferi* are the major etiological agents of this disease, and *S. parasitica* is known as the most important among them (*Van West, 2006*). *S. parasitica* penetrate into epidermal tissues, usually colonizing the tail or head region and then proliferate to cover the entire body surface (*Willoughby, 1994*).

Traditionally, *S. parasitica* infections were effectively controlled with malachite green (*Oláh & Farkas, 1978*; *Srivastava & Srivastava, 1978*; *Alderman, 1985*). However, the compound was banned worldwide in 2002 due to its undesirable effects on animal health (*Van West, 2006*; *Stammati et al., 2005*; *Srivastava, Sinha & Roy, 2004*; *Brock & Bullis, 2001*). Since then, the search for new and effective substances against *Saprolegnia* infections has intensified. Although chemicals including formalin, hydrogen peroxide, sodium chloride (*Rach et al., 2005*; *Barnes, Stephenson & Gabel, 2003*; *Schreier, Rach & Howe, 1996*), copper sulfate (*Straus et al., 2009*), detergents such as bronopol (*Pottinger & Day, 1999*) and ozone (*Forneris et al., 2003*) have been shown to be somewhat effective, none were as effective as malachite green. The use of these types of compounds has led to a number of problems, including the development of fungicide resistance and potentially harmful effects to human health (*Phillips et al., 2008*; *Stammati et al., 2005*). As such, there is still an urgent need to develop new alternatives that are effective in combating mycotic infections, but also safe for fish and the environment (*Khosravi et al., 2012*).

Virkon-S was originally developed by Antec International (Antec International Limited, Sudbury, Suffolk, UK) and launched in 1986 for use in farming and livestock production. It is regarded as one of the most advanced farm disinfectants. It was one of the first oxidative disinfectants to be used on the farm and continues to lead the way in livestock production and farm biosecurity, having been successfully deployed against 500 disease-causing pathogens including viruses, bacteria, and fungi, which cause foot and mouth disease, avian influenza, *Salmonella*, and *Campylobacter* (*Marchetti et al., 2006*; *Hernndez et al., 2000*). Due to its wide range of antimicrobial activity and relative safety, Virkon-S is used by the United Nation's Food and Agriculture Organization and governments worldwide to secure biosafety and strengthen Emergency Disease Control Contingency Planning (http://www.virkon.com/en/products-applications/disinfectants/virkon-s/). In Korea, Virkon-S has been approved as a quasi-drug for animals and was used as a disinfectant for aquaculture facilities in 2016.

Despite the wide spectrum of antimicrobial activity of Virkon-S, there have been no reports of its antifungal activity against *Saprolegnia*. In this study, we tested the antifungal activity of Virkon-S against *S. parasitica in vitro* and *in vivo* to determine the possibility of using this material for the control of saprolegniasis in the future.

The animal protocol used in this study has been reviewed and approved by the Pukyong National-Institutional Animal Care and Use Committee (PKNU-IACUC), which outlines the ethical procedures and scientific care of animals used in studies (Approval Number PKNU-2017-01).

## MATERIAL AND METHODS

### *Saprolegnia parasitica* cultures

*S. parasitica* was purchased from the Korean Collection for Type Cultures (KCTC 46452) and cultured on potato dextrose agar (PDA) at 25 °C. This strain was isolated from a farmed rainbow trout from an aquaculture farm in Wonju, Gangwondo, Korea in 2016 before deposition.

### Fish and rearing conditions

In total, 100 fingerlings of common carp, *Cyprinus carpio*, with an average size and weight of 11.5 ± 1 cm and 17.6 ± 3 g, respectively, were obtained from Namsangju Aquaculture Farm, located in Sangju-si, Gyeongsangbuk-do, Korea. Each fish was examined for infection and acclimated in 450-L rearing tanks at 22 ± 2 °C for 10 days. During the acclimatization period, fish were fed twice daily with a proper diet according to *Ellsaesser & Clem (1986)*.

### Virkon-S

Virkon-S was purchased from Bayer Korea (Seoul, Korea) and 1 kg contained triple salt 500 g, hexametaphosphate 181 g, sodium dodecyl benzene sulphonic acid 150 g, malic acid 100 g, sulphamic acid 50 g, and sodium chloride 15 g. A 10% stock was prepared with distilled water and further dilutions were made when necessary.

### Inhibition of spore germination

The inhibitory effects of Virkon-S on spore germination were tested using two methods. First, the minimum fungicidal concentration (MFC), defined as the lowest concentration of a chemical that prevents germination or the visible growth of spores, was determined as described previously (*Yao et al., 2017*; *Hu et al., 2013*). Spore suspension containing oospores and zoospores was prepared as described as *Yao et al. (2017)*. *S. parasitica* was cultured on PDA at 25 °C for 12 days to induce spores. Approximately 10 ml of distilled water was added to each 87-mm diameter Petri dish and the mycelium and spores were scraped and filtered through eight layers of sterile cheesecloth. The final concentration of spores was adjusted to approximately $1 \times 10^6$ spores/ml using a hemocytometer. A 10 µl sample of spore suspension was spotted on the center of a 87-mm diameter Petri dish containing 20 ml of PDA with 0, 2, 4, 10, 20, or 100 ppm Virkon-S, and incubated at 25 °C. After 72 h of incubation, the diameter of mycelial growth was measured. Inhibition of spore germination was also determined by percent spore germination as described

by *Király et al. (1974)*. PDA plates containing 0, 2, 4, 10, 20, and 100 ppm Virkon-S were prepared but the amount of total PDA was only 10 ml for each plate. This condition reduced mycelial growth and the plate was transparent enough to observe spore germination under a light microscope. Three spots of a 10-µl spore suspension were placed on each plate and incubated at 25 °C for 72 h. The percent spore germination was determined as follows:

$$\text{Percent spore germination} = \frac{\text{No. of spores germinated}}{\text{Total no. of spores examined}} \times 100.$$

## Mycelial growth inhibition on PDA plates

Inhibition of *S. parasitica* mycelial growth was tested on PDA plates containing different concentrations of Virkon-S as described by *Hu et al. (2013)*. Briefly, 2× PDA was sterilized, cooled to approximately 65 °C, and mixed with the same volume of Virkon-S to give final Virkon-S concentrations of 2, 4, 10, 20, and 100 ppm. 2× PDA was mixed with the same volume of sterile water in control plates. Aliquots of the mixture (10 ml) were poured onto 87-mm diameter Petri dishes. A *Saprolegnia*-colonized PDA block of approximately $5 \times 5$ mm was placed on the center of the prepared plates. The plates were incubated at 25 °C for 72 h and the diameter of the mycelial growth was measured. The percentage of fungal inhibition was calculated based on the percent inhibition of radial growth (PIRG) as described by *Dananjaya et al. (2017)* as follows: PIRG (%) = $[(R1 - R2)]/R1 \times 100\%$, where R1 = radial growth in control and R2 = radial growth in treatment.

## Cytotoxicity of Virkon-S in epithelioma papulosum cyprini (EPC) cells

The cytotoxic effect of Virkon-S was evaluated using the Ez-Cytox Cell Viability Assay Kit (Dogen-Bio Co., Ltd., Seoul, Korea) with EPC cells by following the procedures described by *Park et al. (2017)*. EPC cells ($1 \times 10^5$) were grown in L-15 medium supplemented with 10% fetal bovine serum, penicillin (62.5 µg/ml), and streptomycin (100 µg/ml) in 96-well plates overnight. For treatment, the cell medium was replaced with medium (100 µl) containing 10, 100, 500, 1,000, 5,000, and 10,000 ppm of Virkon-S. Non-treated cells were used as a negative control. After 24 h of incubation, 110 µl of medium containing 10 µl of water-soluble tetrazolium solution was added to each well, and the plates were incubated for a further 4 h. The absorbance at 460 nm was measured using an enzyme-linked immunosorbent assay reader (Molecular Devices, Silicon Valley, CA, USA), and relative cell viability was calculated using cells treated with medium only as a control.

## Acute toxicity in common carp

Water-only toxicity tests were carried out with cultured fingerlings of the common carp, *Cyprinus carpio*, using five concentrations of Virkon-S (2, 4, 10, 20, and 100 ppm) and three fish per concentration. Healthy and disease-free fish that were not previously exposed to any pollution agents or toxicants were selected based on their activity and external appearance. The fish were acclimated for 7 days at 18 °C under constant light and feeding with commercial feed. Three fish were placed in a 3-L glass flask containing different concentrations of Virkon-S and kept for 96 h in aerated water without any water changes or feeding. Dead fish were removed from the flask to prevent possible deterioration of the
water quality. The percentage of fish mortality was calculated for each concentration after 24, 48, 72, and 96 h of exposure. After 96 h of exposure, all of the survivors were transferred to a 50-L tank equipped with aeration and observed for 40 days for any post-exposure effects.

### Inhibition of saprolegniasis by Virkon-S

Five groups of 20 fish were used for artificial infection with *S. parasitica* and treatment with Virkon-S. Three groups were artificially infected and treated with 2, 4, and 10 ppm Virkon-S. Fish in the positive control group were infected with *S. parasitica* but not treated with Virkon-S. Fish in the negative control group were neither infected with *S. parasitica* nor treated with Virkon-S. Fish were kept in a 150-L glass fiber tank separated into three sections with a metal screen with six or seven fish in each section. The water temperature was fixed at $17 \pm 1$ °C using a room air conditioner. All fish were observed for behavioral and clinical signs of infection.

For artificial infection with *S. parasitica*, fish were descaled and wounded in three places on one side of the body (top of the head, center of the body, and the tail region) using a sharp scalpel. The wounded area was approximately 1 cm$^2$. Spores of *S. parasitica* were prepared as above and added to tanks containing the three treatment groups and the positive control group at a final concentration of $1 \times 10^5$ spores/ml (*Willoughby, 1994*; *Hatai & Hoshiai, 1994*).

Ten days after the addition of spores, white cotton wool-like growth on the surface of the wounded area was observed with a light microscope and cultured on PDA at 25 °C for 3 days. The morphological and microscopic characteristics of the culture were compared with the characteristics of *S. parasitica* recorded from previous studies (*Hatai, Willoughby & Beakes, 1990*; *Willoughby, Pickering & Johnson, 1984*). Virkon-S solution was added to these treatment tanks at final concentrations of 2, 4, and 10 ppm to assess the treatment effect. The water in the tank was replaced with the same concentration of Virkon-S after 5 days and the cumulative mortality was recorded for 10 days. The fish were diagnosed by clinical signs and lesion healing 10 days after treatment.

## RESULTS

### Inhibition of spore germination

The inhibition of *S. parasitica* spore germination by Virkon-S was tested by observing the mycelial growth from a suspension of spores on PDA plates of normal thickness (20 ml media in 87-mm plates) and enumerating spore germination on PDA plates of half thickness. There was no sign of mycelial growth on plates containing 4, 10, 20, and 100 ppm Virkon-S 72 h after incubation at 25 °C (Figs. 1A–1D). Mycelial growth on a plate containing 2 ppm was 53.8% of the control plate (35 vs. 65 mm, respectively) (Fig. 1E). Therefore, MFC was defined as less than or equal to 4 ppm on PDA plates.

There was no sign of spore germination on plates of half thickness containing 4, 10, 20, or 100 ppm Virkon-S 72 h after incubation at 25 °C. A small number of spores on PDA plates containing 2 ppm germinated at a rate of 39.2% (22 out of 56). In contrast, most of the spores on control plates germinated at a rate of 96.2% (51 out of 53 spores). Therefore,

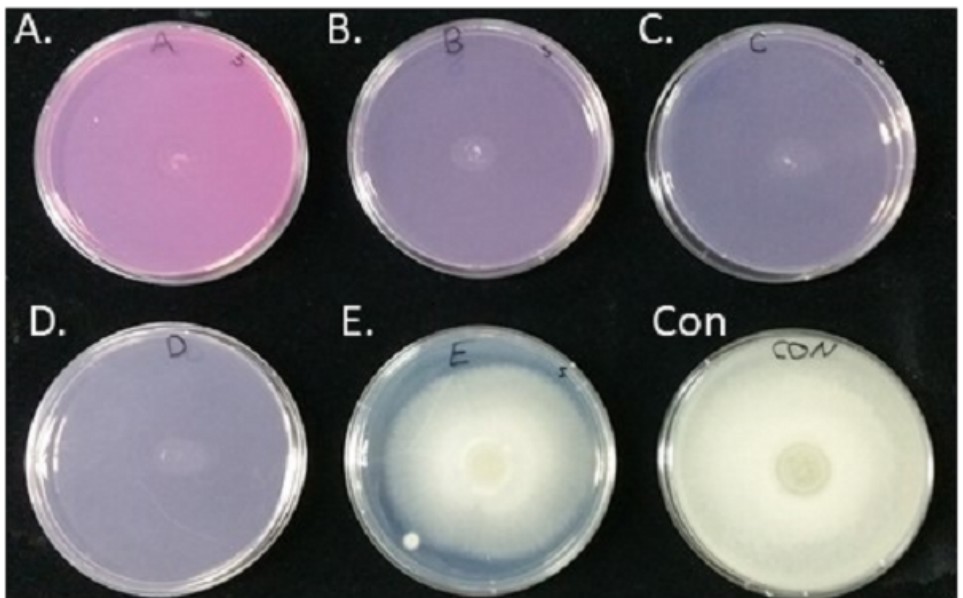

**Figure 1 Inhibition of spore germination and the resulting mycelial growth on potato dextrose agar (PDA) plates using different concentrations of Virkon-S.** The images were taken 72 h after incubation at 25 °C. A; 100 ppm, B; 20 ppm, C; 10 ppm, D; 4 ppm, and E; 2 ppm. No Virkon-S was added in the control plate. Photo was taken by the first author.

the inhibition rate of spore germination was calculated as 59% at a concentration of 2 ppm $[(96.2–39.2)/96. 2\times 100]$.

## Mycelia growth inhibition

The growth of *S. parasitica* on PDA containing 2, 4, 10, 20, and 100 ppm Virkon-S was observed after inoculation at 25 ° C for 72 h. No growth of mycelia was observed in the plates containing 10, 20, and 100 ppm Virkon-S (Figs. 2A–2C). There was slight growth of *S. parasitica* on PDA plates containing 4 ppm Virkon-S (Fig. 2D). The growth inhibition rate (IR) on plates containing 4 and 2 ppm was 69.2 and 25.6%, respectively when the IR was calculated as [%IR = 100 − 100X/Y], where X = mycelia growth in sample; Y = mycelia growth in control.

## Cytotoxicity of Virkon-S in cultured EPC cells

The cytotoxic effect of Virkon-S in EPC cells is shown in Fig. 3. There was no effect of Virkon-S on cell viability at 10 and 100 ppm. There was a slight decrease (97%) at 500 ppm. However, the cell viability dropped suddenly when the Virkon-S concentration was greater than 1,000 ppm. In the mycelial growth inhibition and spore germination inhibition assays, 4 ppm was enough to inhibit spore germination and the resulting mycelial growth. Even a concentration of 100 ppm did not result in any toxicity in EPC cells. Therefore, 10 ppm and lower concentrations were used in further experiments.

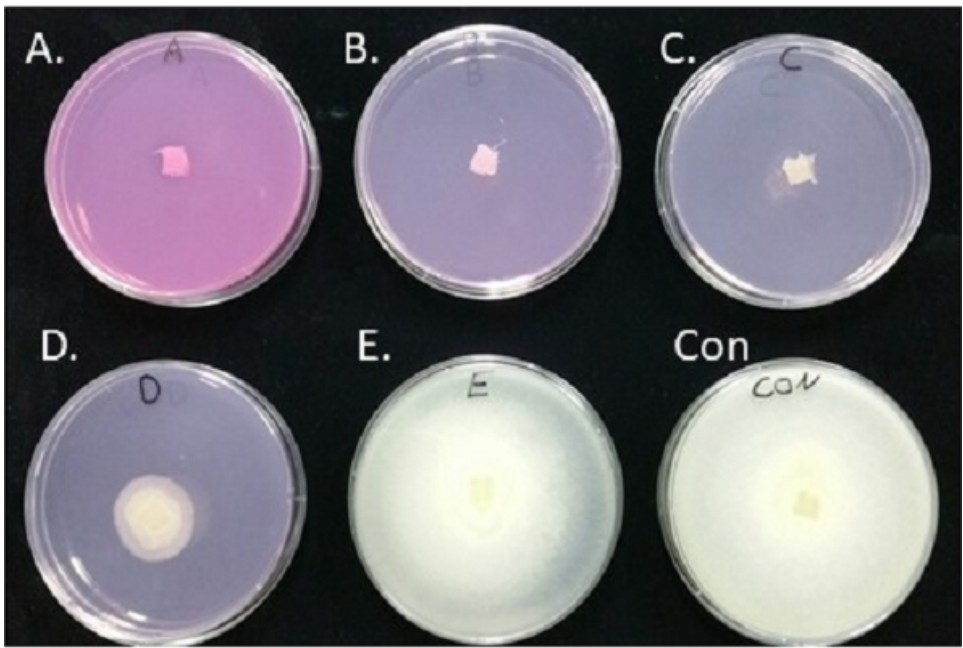

**Figure 2** **Inhibition of *S. parasitica* m ycelia growth on PDA plates containing different concentrations of Virkon-S.** A; 100 ppm, B; 20 ppm, C; 10 ppm, D; 4 ppm, and E; 2 ppm. No Virkon-S was added in the control plate. Photo was taken by the first author.

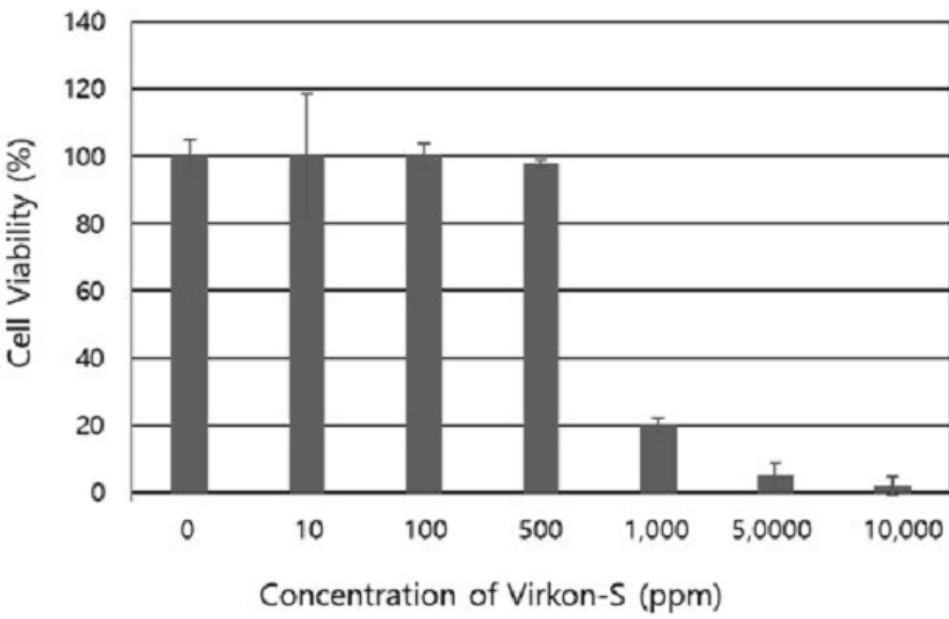

**Figure 3** **Cytotoxic effect of Virkon-S on cultured epithelioma papulosum cyprini cells.** The results represent the average of eight replications and the standard deviations are indicated.

**Table 1  Cumulative mortality of common carp during acute exposure to Virkon-S.**

| Concentration (ppm) | Cumulative mortality (%) | | | |
|---|---|---|---|---|
| | 24 h | 48 h | 72 h | 96 h |
| 100 | 100 | – | – | – |
| 20 | 75 | 100 | – | – |
| 10 | 0 | 0 | 0 | 0 |
| 4 | 0 | 0 | 0 | 0 |
| 2 | 0 | 0 | 0 | 0 |

## Acute toxicity test of Virkon-S in common carp

Acute toxicity of Virkon-S in common carp was investigated by placing fish in glass flasks containing five different concentrations of Virkon-S for up to 96 h. As shown in Table 1, all fish in the 100-ppm flask died within 24 h. Moreover, 75 and 100% cumulative mortality was observed within 24 and 46 h, respectively, in the 20-ppm flask. However, no mortality was observed in the flasks containing 10, 4, and 2 ppm Virkon-S. Furthermore, the survivors from the acute toxicity test did not show any after-exposure effects when they were kept in a culture tank for 40 days. Therefore, it was concluded that 10 ppm, which inhibited spore germination and mycelial growth, did not cause any cytotoxic effects in EPC cells, and therefore can be defined as the maximum acceptable toxicant concentration (MATC) for Virkon-S.

## Induction of artificial infection and treatment with Virkon-S

Seven days after the addition of *S. parasitica* spores to the tanks containing artificially wounded common carp, the typical signs of saprolegniasis, including cotton shape growth of fungi and wound ulceration, appeared (Fig. 4A). The cotton wool-like mycelial growth was removed from the wound and observed with a microscope to confirm its resemblance to the inoculated *S. parasitica*, and placed on a PDA plate to induce mycelial growth. Both the cotton wool-like mycelial growth from the infected fish and mycelia grown on the PDA plate showed the same morphological characteristics of *S. parasitica*, which indicated that the disease was induced by the spores of *S. parasitica* that had been added to the tanks. The diseased fish were treated with Virkon-S at a final concentration of 2, 4, and 10 ppm and the results are shown in Table 2. During the 10-day treatment period, no mortality was observed among fish treated with 4 and 10 ppm. However, the cumulative mortality in the 2-ppm tank reached 50%. Furthermore, fish treated with 4 and 10 ppm showed clear recovery from the disease. As shown in Fig. 4B, all of the cotton shape fungal growth disappeared from the body of the fish. Furthermore, new scales appeared and covered the wounded area, which indicated complete recovery from the disease.

## DISCUSSION

Since malachite green has been banned for the treatment of saprolegniasis due to possible genotoxicity, carcinogenetic effects, and residual toxicity, this disease has resulted in severe economic losses in the fresh water fish farming industry (*Van West, 2006*). Although many

A.

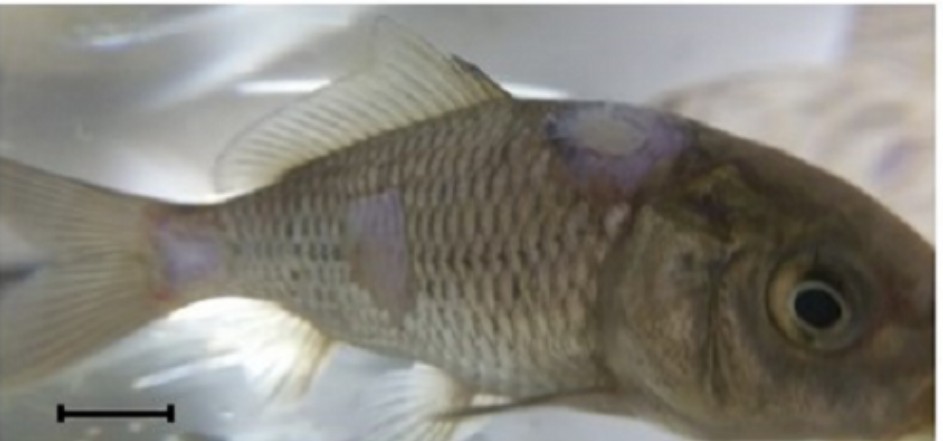

B.

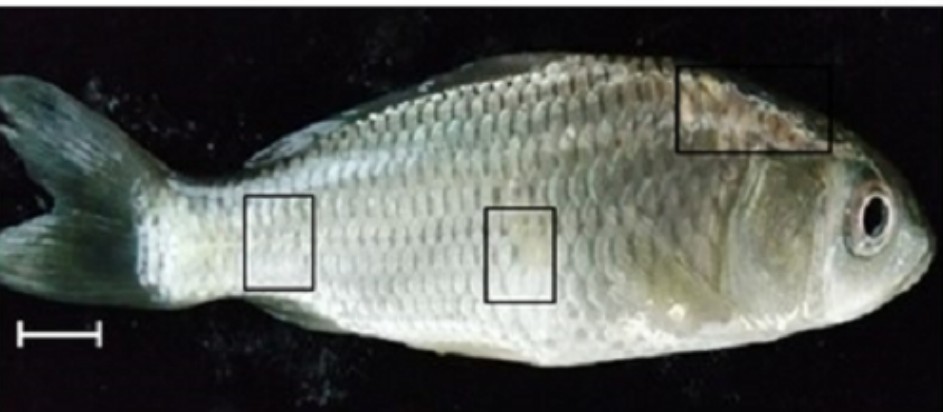

**Figure 4** **Artificial induction of saprolegniasis in common carp and treatment with Virkon-S.** (A) Clinical signs of *S parasitica* on the fish body 10 days after artificial wound induction and exposure to the fungal zoospore. (B) Recovery from saprolegniasis after treatment for 10 days with 10 ppm Virkon-S. The rectangles indicate the wounded area for artificial infection and show complete recovery from the disease. Scale bars indicate 1 cm. Photos were taken by the first author.

alternatives have been tried, no chemicals are presently available that provide sufficient protection against the disease. In addition, sanitary problems, environmental restrictions, and high cost have also limited the use of these synthetic antimicrobials (*Yao et al., 2010*). The only US Food and Drug Administration-approved compounds for fungus control are 37% formalin and 35% hydrogen peroxide, but the efficiency is lower than that of malachite green (*Straus et al., 2016*).

Virkon-S is a well-known disinfectant that has been proven to be effective against bacteria, viruses, and fungi (*Gehan et al., 2009*). In this study, we showed that Virkon-S has

**Table 2  Healing of artificially induced saprolegniasis by Virkon-S treatment.**

| Treated concentration (ppm) | % of healing after treatment | Cumulative mortality (%) |
| --- | --- | --- |
| 10 | 100 | 0 |
| 4 | 100 | 0 |
| 2 | 50 | 0 |
| Positive control[a] | 0 | 90 |
| Negative control[b] | – | 0 |

Notes.

[a] Positive control group was artificially infected with *S. parasitica* spores but was not treated with Virkon-S.

[b] There was neither artificial infection nor treatment with Virkon-S.

antifungal activity against *S. parasitica in vitro* and *in vivo*, and to the best of our knowledge, this is the first time this activity has been demonstrated.

The minimum concentration for inhibition of spore germination and the resulting mycelial growth on PDA plates was as low as 4 ppm, and partial (53.8%) inhibition was observed on a PDA plate containing 2 ppm Virkon-S. Therefore, the MFC was determined to be 4 ppm on PDA plates. When the inhibition of spore germination experiment was repeated on thin PDA plates for enumeration, complete spore germination inhibition still occurred at 4 ppm and 59% inhibition at 2 ppm was obtained. Although a concentration of Virkon-S as low as 4 ppm was proven to be effective to inhibit germination in our experiment, the concentration and contact time required for fungi inactivation seems to be specific to each fungus. It has been reported that 1 min of contact with 1% Virkon-S is sufficient to inactivate *Batrachochytrium dendrobatidis*, which causes the mass mortality of various amphibian species (*Gold et al., 2013*; *Johnson et al., 2003*). On the contrary, *Gehan et al. (2009)* reported that *Aspergillus fumigatus* and *Fusarium* species were resistant to 1% Virkon-S solution following 30 min and even 60 min of contact in the presence of organic materials.

Similarly, *Rogawansamy et al. (2015)* reported that a 10% concentration of Virkon-S resulted in a mean inhibition zone diameter of only 19.25 mm ($\pm$7.08) for *A. fumigatus*, and 18.67 mm ($\pm$1.15) for *Penicillium chrysogenum*. Furthermore, 5, 3, and 1% Virkon-S had no effect on the growth of either fungi. Therefore, it seems that the anti-fungal effect of Virkon-S against each target fungus needs to be tested for practical application of Virkon-S.

To use Virkon-S for the control of saprolegniasis, it should be safe for fish or treated embryos. The cytotoxicity of Virkon-S was tested with EPC cells, which originated from the fathead minnow, *Pimephales promelas*. No cytotoxic effect was observed at 10 and 100 ppm, although there was a slight decrease in cell viability (97%) at 500 ppm (Fig. 3). There have been no reports of the cytotoxicity of Virkon-S in cultured fish cell lines. However, its ability to inactivate animal viruses has been tested on animal cells. For example, *Wu et al. (2017)* reported that the cytotoxicity of 1% Virkon-S on cultured baby hamster kidney 21A cells was completely abolished at a 1 in 81 dilution (123 ppm) in cell culture media, which is similar to our data and significantly lower than the recommended 1% concentration for use as a disinfectant.

Our results indicated that 4 ppm was sufficient to inhibit spore germination and mycelial growth. Therefore, it seems that Virkon-S could be used to control the growth of *S. parasitica* without any cytotoxic effects at a concentration that can inhibit the fungus. This was further confirmed by acute toxicity tests. No toxic effect was observed in fish kept at 2, 4, and 10 ppm for 96 h (Table 2). Moreover, these fish did not show any after-exposure effects up to 40 days after termination of the test. Considering the results from all of the experiments, the MATC was determined to be 10 ppm.

The efficacy of Virkon-S was tested with fish that were artificially infected with *S. parasitica*. Fish that were wounded and inoculated with *S. parasitica* spores showed clear symptoms of saprolegniasis 10 days after inoculation, which was confirmed by microscopic observation and colony morphology on PDA plates. Fish treated with 2, 4, and 10 ppm Virkon-S showed 50, 100 and 100% disease recovery, respectively after 10 days of treatment. The concentrations required for the control of the disease coincided with the concentrations obtained from the *in vitro* assay. In addition to 100% survival of the treated fish, all recovered fish showed regeneration of scales on the wounded area, which is an indication of successful treatment.

## CONCLUSIONS

Virkon-S is approved as a quasi-drug that can be used as a disinfectant for the treatment of aquaculture facilities and equipment in Korea. The results of this study indicate that 4 ppm of Virkon-S can also be used for the control of saprolegniasis with no toxic effects on cultured fish cells or fish in tanks. Although further research on the effect in humans is necessary, Virkon-S is a good candidate for the control of saprolegniasis, which causes huge economic losses in the aquaculture industry.

### Funding

This work was supported by a Research Grant from Pukyong National University (year 2017). The funders had no role in study design, data collection and analysis, decision to publish, or preparation of the manuscript.

### Grant Disclosures

The following grant information was disclosed by the authors:
Research Grant from Pukyong National University.

### Competing Interests

The authors declare there are no competing interests.

### Author Contributions

- Haitham Saeed Rahman conceived and designed the experiments, performed the experiments, analyzed the data, contributed reagents/materials/analysis tools, prepared figures and/or tables, authored or reviewed drafts of the paper, approved the final draft.

- Tae-Jin Choi conceived and designed the experiments, contributed reagents/materials/-analysis tools, prepared figures and/or tables, authored or reviewed drafts of the paper, approved the final draft, getting the permit for using fish for experiment.

### Animal Ethics

The following information was supplied relating to ethical approvals (i.e., approving body and any reference numbers):

The animal protocol used in this study has been reviewed and approved by the Pukyong National-Institutional Animal Care and Use Committee (PKNU-IACUC), which outlines the ethical procedures and scientific care of animals used in studies (Approval Number PKNU-2017-01).

### Data Availability

The raw data are provided in a Supplemental File.

### Supplemental Information

Supplemental information for this article can be found online at http://dx.doi.org/10.7717/peerj.5706#supplemental-information.

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
