# Peer review of "The efficacy of Virkon-S for the control of saprolegniasis in common carp, Cyprinus carpio L"

_PeerJ, doi:10.7717/peerj.5706_

## Round 0.1 · original submission · Major Revisions

Both reviewers have clearly established several suggestions and comments about your manuscript, so no additional information is required by the editor at this stage. Please, follow their suggestions and indicate the changes you have made in the text.

Thank you for your understanding.

Reviewer 1 ·

Basic reporting

This manuscript describing the safety and applicability of Virkon-S for treatment of saprolegniasis in common carp is well-written and provides sufficient background outlining the problem being studied.
Several items which may improve clarity are listed below in order of occurrence:
1. Credit line (line 7) should include country of origin.
2. Lines 56-58 and throughout manuscript and reference list: all formal latinized taxon names (phylum, class, order, family, genus, species) should be written in italics.
3. Lines 57-59: make the definition of “Saprolegnia complex” and its role in disease clearer. Suggest introducing the two families within the order Saprolegniales before introducing the main genera.
4. Line 85: reword to avoid using the phrase “secure biosecurity”.
5. Line 106: reverse numbers to coincide with average size and weight.
6. Line 109: was “apparent satiation” a measurable status? If so, how was it objectively measured? If not, this phrase can be deleted.
7. Line189: after “added to” might insert “tanks containing”
8. Line 234: no need to state “and 4 ppm is even lower than 100 ppm”.
9. Line 291: change “flagella” to “flagellated”.
10. Line 301: word, “species” should not be in italics.
11. Figure 2: significant structures as described in the text are difficult to see in the figure ( germinated, un-germinated, and bulged spores). It may be useful to point these out in the figure.

Experimental design

The research question was clearly-defined and addressed an important knowledge gap. The investigation was rigorous, and of high technical and ethical standards. Method details were sufficient to allow replication by other investigators.

Validity of the findings

Appropriate controls were included and data is statistically sound. Conclusions are appropriately linked to the original research question and provide information that is immediately applicable to the aquaculture industry.

Reviewer 2 ·

Basic reporting

Strengths: The manuscript presents an important fish disease and a potential treatment. The experimental plans are very well design, especially the toxicity assay. The results from these parts meet publication quality.

Weaknesses:
• Several sentences are not clear and should require more rigorous professional English to conform to professional standards.
• Please provide numbers of fish and replicates used in the experiments
• The quality of figure 2 suffers from a good resolution and clarity.
• Raw data of toxicity in fish experiment was not shared.

Experimental design

I have a major concern with the spore inhibition assay. Figure 2 is definitely not zoospores. This entire experiment needs to be re-investigate.

Line 108: acclimatization not acclimation
Line 121: Growth on PDA for 12 days would not induce zoospores. In fact, young mycelium 3-5 days should be used to produce zoopsores. The induction of zoospore production is triggered by starvation. In the lab setting, the sporangia induction and release of zoopspores are done by flooding.
Line 122: Please indicate, how long were the plates incubated in distilled water?
Line 124: Misspelled “concentration”
Line 127: Please provide the parameter used whenviewing with naked eye. Surface area?
Line 127: Replace “mycelia growth” with “mycelial growth”
Line 139: Replace “mycelia growth” with “mycelial growth”
Line 156: For consistency, please use the same units for antibiotic concentrations.
Line 171 and 172: Pleae clarify. How long were the fish placed in Virkon-S? How long are they in aerated water?
Lines 195 to 198: Sentences were not clear. When was Virkon-S added? 10 days after zoospore inoculation? This experimental method needs revision.

Validity of the findings

The entire experiment of spore inhibition assay needs to be re-investigate.
Fig. 3A-C: Why plate A is pink while others aren’t. Plate D looks like it has a contamination (probably replace it with a better picture)
Fig 2: with new data, zoospores and sporangia should be labelled.
Line 208: It doesn’t look like 2ppm plate is 53.8% of 4ppm. Also, 4ppm plate appears to have contamination.
Line 211: What is meant by bulging of spores? Need to show (label) that in the pictures. Also, why should bulging happen? Is that an effect of Virkon S?
Line 223: Growth in 4D doesn’t seem ‘slight’.
Line 252 and 254 : Replace “cotton wool” with “mycelial growth” or “cotton wool-like structure”.
Line 290 to 294: These sentences are not clear. “Most of the structures observed in the samples treated with Virkon-S were not flagella. Inhibition of cyst germination by several chemicals has been reported from the well-known plant pathogenic oomycetes, Phytophthora infestance (Matheron et al., 2000). Thus, it is possible that these bulged structures were cysts that formed from the primary zoospores”

Line 291: At this magnification, it may not be possible to view the flagella.
Line 292. Misspelled “infestans”
Line 327: “colony morphology on PDA” instead of “culture on PDA plates”.
Iine 331: Use “successful treatment” instead of “complete treatment”

Additional comments

Overall, this is an interesting study, but requires professional English writing. My major concern is around the spore inhibition assay. I do not think the authors used zoospores, probably oospores, I could not tell but definitely not zoospores. The authors should pay particular attention to ensuring that the methods and figures are complete and that new results are replaced and presented in the discussion.

---

## Round 0.2 · Minor Revisions

Dear author,
The manuscript will be accepted for publication after addressing some last suggestions proposed by one of the reviewers.
Thank you very much for your cooperation.

Reviewer 1 ·

Basic reporting

This manuscript describes an important observation regarding the potential application of Vircon-S for treatment of saprolegniasis in fish. The authors have outlined the background sufficiently and the manuscript is generally well-written.
I believe that two items, not directly related to the main point of this study, should be addressed. The first is use of the word, fungus, to describe Saprolegnia. Oomycetes are not considered true fungi; use of the word fungus should at least be explained in the introduction. Secondly, the issue of spore type (zoospore or oospore) used in the study or inability to distinguish should be addressed.
Specific points
1. line 60; remove “only”
2. lines 63-67; This lengthy sentence is not clear. It contains three points, 1) all species of Saprolegnia responsible for saprolegniasis have not yet been identified – what is this statement based on? Is it significant to this study? 2) Saprolegnia parasitica and Saprolegnia diclina are difficult to distinguish and are often referred to as the Saprolegnia complex. 3) members of the Saprolegnia complex and Achlya hoferi are the major agents that have been associated with saprolegniasis and, among them, S. parasitica is the most important (frequent?) How does major and most important differ?
3. line 77; remove “and”
4. line 78; remove first “and”
5. line 82; remove first semi-colon in parenthesis
6. line 138; change “spores suspension” to suspension of spores or spore suspensions
7. line 168; change “mycelia” to mycelial growth
8. line 219; remove secone “white”
9. lines 236-237; There were four concentrations but only three images in Fig. 1A-C.
10. line 239; should be MFC and less than or equal to 4 ppm
11. line 253; What is in plate2E?
12. line 255; X/Y = diameter of mycelial growth
13. lines 276-277; ….which inhibited spore germination…
14. line 286; change “that it was indeed” to ‘its resemblance to’
15. line 292; remove, change or verify use of the word, zoospore, in Table 2

Experimental design

The design and technical conduct is of high quality and reproducible.

Validity of the findings

The findings included appropriate controls and are valid.

---

## Round 0.3 · accepted · Accept

Many authors consider saprolegniasis as the most relevant mycotic disease affecting both wild and culture fish. This manuscript supposes a great advance in the field and will be very useful for the control of this disease.

#